# The Influence of Vaginal Native Tissue Repair (VNTR) on Various Aspects of Quality of Life in Women with Symptomatic Pelvic Organ Prolapse—A Prospective Cohort Study

**DOI:** 10.3390/jcm9061634

**Published:** 2020-05-28

**Authors:** Ewa Rechberger, Katarzyna Skorupska, Tomasz Rechberger, Aleksandra Kołodyńska, Paweł Miotła, Beata Kulik-Rechberger, Andrzej Wróbel

**Affiliations:** 1Second Department of Gynecology, Medical University of Lublin, Jaczewskiego 8, 20-954 Lublin, Poland; ewarechberger92@gmail.com (E.R.); rechbergt@yahoo.com (T.R.); kolodynska.aleksandra@gmail.com (A.K.); pmiotla@wp.pl (P.M.); wrobelandrzej@yahoo.com (A.W.); 2Department of Paediatric Propedeutics, Medical University of Lublin, Gębali 6, 20-093 Lublin, Poland; brechberger@interia.pl

**Keywords:** urinary incontinence, pelvic organ prolapse, stool incontinence, vaginal native tissue repair, quality of life

## Abstract

Pelvic organ prolapse (POP) and the associated functional disorders are a major epidemiological problem that compromises the quality of life (QoL). The aim of this study was to assess the impact of lower urinary tract symptoms (LUTS) related to POP and vaginal native tissue repair (VNTR) on QoL. Two hundred patients with symptomatic POP were stratified into four groups according to the dominant storage phase function disorders: Urgency; stress urinary incontinence (SUI); mixed urinary incontinence (MUI), and without clinically significant symptoms from lower urinary tract (LUT). They underwent VNTR from January 2018 to February 2019. After 12 months, the QoL was assessed by the Prolapse Quality of Life (P-QoL) and visual analogue scale (VAS) questionnaires. The data were analyzed with Statistica package version 12.0 (StatSoft, Krakow, Poland), using the Kalmogorow–Smirnoff, Shapiro–Wilk W and the one-way analysis of variance with post hoc Tukey tests. The results of P-QoL showed significant improvement (*p <* 0.05) in all the study groups in most domains assessed before surgery and 12 months after surgery. Significant improvements in all the symptoms assessed by the VAS scale results were found in groups Urgency and MUI. The LUTS questionnaire revealed significant improvement in all voiding and post voiding symptoms in these groups. VNTR effectively eliminated LUTS and significantly improved the patients’ QoL associated with POP.

## 1. Introduction

Pelvic organ prolapse (POP) and the accompanying functional disorders are significant epidemiological, medical and social problems. About 50% of all women will develop some degree of prolapse during their life, but very often the anatomical changes correspond neither to the severity of the prolapse nor the symptoms associated with it. Depending on the methods of screening contemporary reports, the prevalence of serious POP is estimated to be between 3% and 6% [1]. The main factor in the etiology of POP is the weakness of the endopelvic fascia and the muscles of the pelvic floor. This may result in the herniation of the pelvic organs leading to prolapse [2]. There is a bulk of evidence that anatomical lesions of the levator ani muscle play an important role in the etiology of pelvic organ prolapse. There is sufficient evidence to state that delivery-related levator trauma is associated with the objective prolapse in clinical assessment especially of the anterior and central compartments, with an odds ratio of 5.7 (95% CI, 3.5–9.1) [3]. Moreover, there is also an association between urinary incontinence (UI), fecal incontinence, pain during sexual intercourse (and hence, avoidance) in patients affected by POP, although this latter correlation is not clearly and unanimously confirmed [4].

Symptomatic POP refractory to the conservative management of either pelvic floor muscle training or vaginal pessaries may warrant surgical intervention. Currently, because several global regulatory agencies have issued health warnings concerning mesh products due to increased complications such as erosion, infection, chronic pain and sexual problems, surgical treatment strategies have returned to being the native primary repair approach.

Vaginal native tissue repair (VNTR) is in most of the cases the first treatment option of POP that is recommended by the International Continence Society/International Urogynecological Association (ICS/IUGA) committees. The intent of applying VNTR is to bring about the restoration of the normal anatomy, the prevention of recurrences and finally the relief of patient’s symptoms [5].

The visual analogue scale (VAS) is a means of measuring the subjective satisfaction declared by patients. The VAS was originally developed for pain assessment and has been incorporated into various urogynecological symptom evaluations that facilitate the quick assessment of the patient’s subjective bother related to certain symptoms [6,7]. The Prolapse Quality of Life questionnaire (P-QoL) is an objective tool for assessing the quality of life (QoL) in patients with POP [8].

The aim of this study was to assess objectively and subjectively the impact of VNTR on quality of life, pain, urine incontinence, fecal incontinence and sexual functions related to POP.

## 2. Materials and Methods

This prospective cohort study involved 200 patients who underwent VNTR due to POP (the second or higher degree according to POP-Q scale) in a single high-volume Gynecological Centre, from January 2018 to February 2019. None of the patients were previously operated due to POP. While taking their detailed medical history, all the eligible participants were asked before and after surgery for the occurrence of common lower urinary tract symptoms (LUTS) (as previously described), and the Patient Global Impression of Severity was applied (with response options 0: not present; 1: very mild; 2: mild; 3: moderate; 4: strong; 5: very strong) in order to assess how annoying for the patients was their storage, voiding and post micturition symptoms [9,10].

The participants of the study were placed into four groups depending on the urinary storage phase disorder: group 1: with urgency; group 2: with stress urinary incontinence; group 3: with mixed urinary incontinence, and group 4: without clinically relevant storage phase symptoms. Voiding symptoms (slow stream, splitting or spraying, intermittent stream, hesitancy, straining, terminal dribbling) and post-micturition symptoms (feeling of incomplete emptying and post micturition dribbling) were additionally assessed in all patients.

The degree of POP was measured by way of applying the POP-Q scale [11]. The observation time was 12 months. Before the procedure, as well as 6 weeks, 6 and 12 months afterwards, all the patients underwent repeated detailed clinical assessment, and their QoL were assessed by the VAS (subjectively) and P-QoL (objectively) questionnaires. The ten-point VAS scale (0: no distress; 2: annoying; 4: uncomfortable; 6: dreadful; 8: horrible and 10: unbearable distress) embraced five questions in order to assess subjectively: 1: pain related to POP; 2: impact of UI on functioning in everyday life; 3: impact of discomfort associated with gas and stool incontinence on everyday life; 4: vaginal discomfort on functioning in everyday life; and 5: the impact of POP on sexual functions [12].

The P-QoL is a disease-specific QoL questionnaire that has proven to be a valid and reliable instrument for assessing symptom severity, QoL and the treatment outcomes of women with POP. It consists of 38 questions that address nine aspects of everyday life (overall health assessment, impact of POP on QoL, limitations in daily activity, physical/social restrictions, personal relationships, emotions, sleep/energy, severity of disorders and symptoms).

Before inclusion, all patients gave written informed consent for the participation in the study. The study protocol was approved by the Local Ethics Committee (KE-0254/75). After the eligibility of the study participants was confirmed, they underwent the VNTR procedures as previously described [10]. All the patients, besides supplying detailed written information concerning the type and possible end results of the surgery, had the possibility to maintain everyday phone contact with the first author (ER) so as to gain additional psychological and medical support if necessary.

The statistical required sample size for the effect size f = 0.25 (medium effect size convention) and 0.8 power of the study was computed using the ANOVA a priori test (G*Power, Düsseldorf, Germany). The size of the sample was estimated at 180 [13].

The data were assessed with the Statistica package version 12.0 (StatSoft, Poland), using the Kalmogorow–Smirnoff and Shapiro–Wilk W tests. Furthermore, one-way analysis of variance (ANOVA) was applied, followed by the Tukey post hoc test. A *p* value < 0.05 was considered as statistically significant with 95% confidence.

## 3. Results

Based on the dominant storage lower urinary tract symptoms (LUTS), we ascertained that before the surgery, 55 patients had predominant urgency, 20 had predominantly stress urinary incontinence (SUI), 72 had mixed urinary incontinence (MUI) and 53 did not have clinically relevant symptoms from the lower urinary tract. The POP-Q assessment revealed 58 (29%) women at stage II, 117 (58.5%) at stage III and 25 (12.5%) at stage IV without significant differences in the distribution among the study groups [10]. The demographic data of the patients are presented in Table 1.

The P-QoL results show that the clinically significant improvement occurred in all the study groups in the following domains: general health perception, prolapse impact, role limitation, physical limitation, emotions, sleep/energy, severity measures between time point before the surgery and 6 months after surgery, as well as between the time point before surgery and 12 months after surgery. In the domain of social limitations, there was clinically significant improvement in patients from groups Urgency and MUI. In the domain of personal relationships, we also observed significant improvement in patients from all groups between the time point before surgery vs. 12 months after surgery (Table 2).

The comparisons before surgery vs. 6 weeks after surgery show the clinically significant improvement in all the study groups in the following domains: general health perception and severity measures. We did not, however, observe significant changes after 6 weeks in patients from the Urgency group in the following domains: prolapse impact and personal relationships. In contrast, patients with predominant SUI did not declare clinically significant improvement after 6 weeks in the following domains: prolapse impact, role limitations, physical limitations, social limitations, personal relationships, emotions and sleep/energy. Six weeks after surgery the patients from the MUI group did not show significant change only in the domain personal relationships. Finally, patients without serious symptoms from the lower urinary tract (LUT) did not declare clinically significant changes 6 weeks after surgery when compared to the period before the operation in the following domains: role limitations, social limitations, physical limitations and personal relationships (Table 2).

We found a clinically significant improvement in the VAS scale results (in all the study questions) between the period before the surgery as compared to the 6 and 12 months after in patients from the Urgency and MUI groups (Figure 1, Figure 2, Figure 3, Figure 4 and Figure 5).

The results of the VAS scale clearly indicated that the patients from the MUI group had the most bother in all the studied dimensions and that the VNTR significantly improved the annoying symptoms. The patients from the SUI group declared clinically significant improvement in the VAS scale results in all the questions evaluated after 12 months when compared to the time before surgery, but in comparison, the period 6 months after surgery revealed a positive significant trend only in questions assessing pain related to POP, vaginal discomfort on functioning in everyday life and the impact of POP on sexual functions. Patients without clinically meaningful storage phase symptoms declared significant improvement in the VAS scale results 12 months after the surgery in their replies to all the questions except the question assessing the impact of UI on functioning in everyday life. In this group, the VAS scale assessment after 6 months revealed improvement similarly to the patients from the SUI group. The results of six weeks assessed by the VAS scale in the Urgency and MUI groups showed clinically significant improvement in almost all the questions except for the question assessing the influence of the POP on sexual life. Patients with SUI declared clinically significant improvement after 6 weeks in their replies to questions 1 and 4 similarly to patients without serious symptoms from LUT (Figure 1, Figure 2, Figure 3, Figure 4 and Figure 5).

Statistical analysis performed 6 weeks, 6 months and 12 months after surgery in women from the Urgency group showed significant improvement in all the voiding and post voiding symptoms assessed by the LUTS questionnaire. These improvements (especially the subjective assessment of urine flow and the positive significant influence on splitting during micturition) were quite evident upon comparing the results before surgery to all the time points afterwards. These patients were also less likely to have difficulty when starting micturition and had less urge pressure and less frequent final dribbling throughout study period (Table 3).

Similar results were obtained in MUI group. Statistical analysis performed in this group at 6 weeks and 12 months after surgery showed significant improvement in all the voiding and post-voiding symptoms assessed by the LUTS questionnaire, but improvement in symptoms such as intermittent stream and incomplete emptying the bladder was not seen until 6 months after surgery. Furthermore, between 6 weeks, 6 months and 12 months after surgery, there were no statistically significant differences in the voiding and post-voiding phase symptoms in the examined group of women (Table 3).

In the women from SUI group, the only significant improvement was observed between all the analyzed time points in the incomplete emptying of the bladder (Table 3). There were no statistically significant differences in the voiding phase between the analyzed periods in the group of patients without LUTS symptoms. The analysis performed 6 weeks after surgery, however, showed a significant reduction in the feeling of incomplete bladder emptying in this group of patients (Table 3).

## 4. Discussion

The results of our study, mainly focusing on patients’ quality of life, show that VNTR can be an efficient and safe treatment option for those patients who, beyond POP symptoms, are also suffering from several others functional urogenital tract disorders. Indeed, the results of the P-QoL questionnaire clearly show the reduction of symptoms in the domain of the prolapse impact in all study groups regardless of other complaints. Accordingly, patients with predominant urgency who underwent VNTR declared clinically significant improvement 6 months after surgery, although this result was not seen as early as 6 weeks after surgery. Interestingly, patients from the Urgency and MUI study groups were the ones who benefited most in terms of the significant reduction of LUTS related to POP and this confirms the previous data that urgency was regarded subjectively as the most distressful. Indeed, at 6 and 12 months after VNTR, the patients from both these groups declared improvements in all the aspects of QoL measurable by the P-QoL and VAS scale as well. On the other hand, patients predominantly from SUI group declared improvement 12 months after VNTR as measured by P-QoL in all the domains of the questionnaire, but not in the subjective assessment by VAS scale.

The clinical manifestation of the POP symptoms definitely has a detrimental impact on women’s functional performance and QoL, causing physical, social, psychological, occupational, domestic and/or sexual limitations and alterations of their daily life [14]. There is a bulk of evidence that adequate vaginal apex support, together with correction of the anterior and posterior walls, is essential for a durable effect of surgery in women with advanced prolapse [15,16]. Therefore, all patients included in this study also underwent apical repair as previously described [17]. This technique enables the reconstruction of DeLancey level I and posterior level II, resulting in a physiological width along the whole vaginal axis and symmetrical vaginal lifting with a concomitant reduction of the risk of dyspareunia. The results of our study confirmed that patients suffering from symptomatic POP very often declare many different accompanying symptoms like various kinds of urinary and fecal incontinence or sexual dysfunctions that seriously compromise their QoL. Those bothersome symptoms can be effectively reduced by a number of surgical treatments, bearing in mind that the surgical strategy (if chosen) depends on the degree of POP, functional issues, patient’s age, comorbidities and also on the surgeon’s preferences defined as practice pattern variation (differences in surgical treatment of POP that cannot be explained solely by the underlying clinical condition) and of course, individual surgical skills [18].

A majority of clinicians agree that current surgical strategy should, first of all, take into account the safety of the procedure and the reversibility of the potential complications—and not only the risk of recurrence (as it was in the last years when polypropylene grafts were widely used) [19]. Considering the above, it is clear that VNTR techniques, although probably less effective anatomically (especially in anterior compartment) and of course, not completely free of surgical risks, are much safer for the patients because of the obvious lack of complications related to non-resorbable prostheses such as erosions, extrusions and protracted pain [20].

Our findings are in agreement with the previously published data that OAB symptoms occurring in patients with symptomatic POP could be effectively cured by pelvic reconstructive surgery [21,22]. The results of the LUTS questionnaire show that patients from the Urgency and MUI groups who underwent VNTR had significant improvement, not only in the storage phase disorders, but also pertaining to voiding and post-voiding symptoms. Additionally, the results of our study once more confirm that anterior colporraphy is not an effective treatment choice for SUI. In contrast, Ugianskiene et al. reported that almost half of the patients with SUI before POP surgery became completely dry after prolapse surgery alone. This clearly indicates that not only the differences in surgical technique but also underlying pathophysiology may play pivotal roles in the final functional and anatomical outcomes of prolapse surgery and may explain contradictory results [23].

It has previously been shown that VNTR in the posterior compartment is the preferred approach to rectocele repair because there was no evidence of benefit from the use of mesh inlay or biological graft compared with standard repair in terms of efficacy [24,25]. This was also the case in our whole study group.

Taking into account the reliable scientific data that underlines that POP exerts significant negative impact on the subjective well-being of affected individuals, this assessment should become an integral part of the therapeutic process of women before and after surgical repair of POP in order to ensure a more adequate physical and functional rehabilitation [26,27].

In fact, our study provides an additional proof to the merits of the FDA recommendation that native tissue surgery should be definitely a first surgical therapeutic approach as it results in the significant improvement of women QoL [28,29]. In the present study, the P-QoL and VAS scale clearly point toward a reduction of severity of all the analyzed symptoms at the median follow-up of 12 months.

The strength of our study is its prospective design and large cohort of women who underwent VNTR in the study group, and the fact that we followed the present ICS/IUGA guidelines for surgical outcomes using already validated tools. Among these were the questionnaires for prolapse symptoms. Moreover, all the surgical procedures were done by experienced high-volume surgeons who almost exclude the committing of technical errors during their surgical routines. The main limitation of the study was the single setting and lack of a control group. This last limitation was due to the fact that all our patients were operated due to POP for the first time and according to current international guidelines, we were obliged to use VNTR as the first choice option.

According to our findings, the first recommended surgical strategy for patients with symptomatic POP should be the low-cost VNTR technique without the use of mesh, but with the precise knowledge of pelvic anatomy and fascial compartment that is accompanied by the clinical surgical experience that is mandatory for an optimal clinical outcome.

## 5. Conclusions

VNTR is an effective, safe and durable surgical method for improving the symptoms and quality of life of the patients with symptomatic pelvic organ prolapse. The present study evaluates the short-term effect of vaginal native tissue repair (VNTR) on quality of life. A longer observation time is necessary to conclude the final clinical outcome.

## Figures and Tables

**Figure 1 jcm-09-01634-f001:**
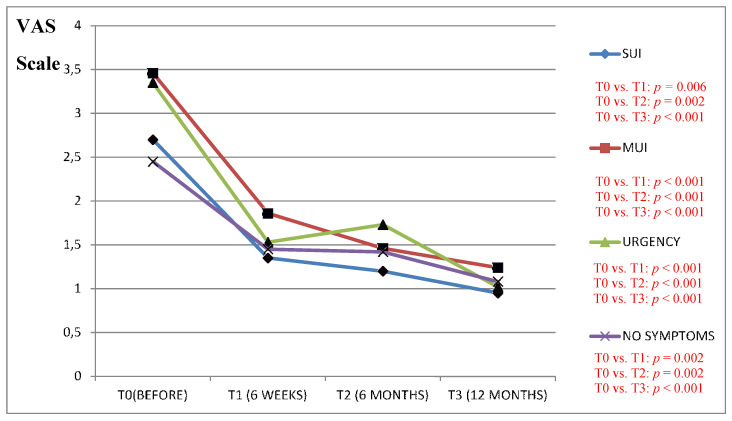
Impact of the vaginal native tissue repair (VNTR) on the pelvic pain associated with pelvic organ prolapse—according to the visual analogue scale (VAS).

**Figure 2 jcm-09-01634-f002:**
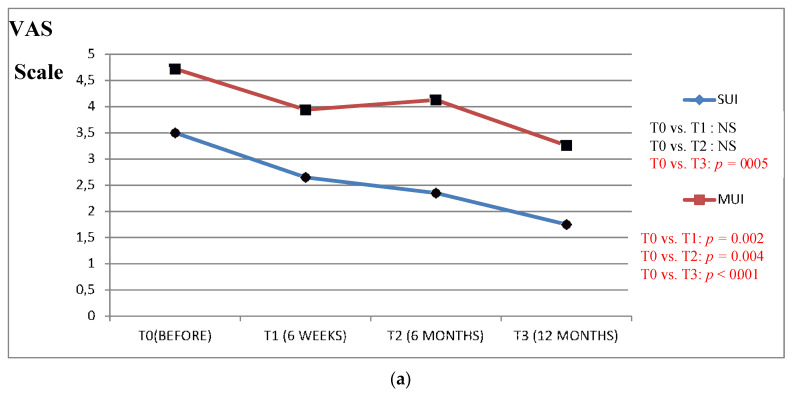
(**a**) Impact of the vaginal native tissue repair (VNTR) on urinary incontinence severity in everyday functioning—according to the visual analogue scale (VAS). (**b**) Impact of the vaginal native tissue repair (VNTR) on urgency severity in everyday functioning—according to the visual analogue scale (VAS).

**Figure 3 jcm-09-01634-f003:**
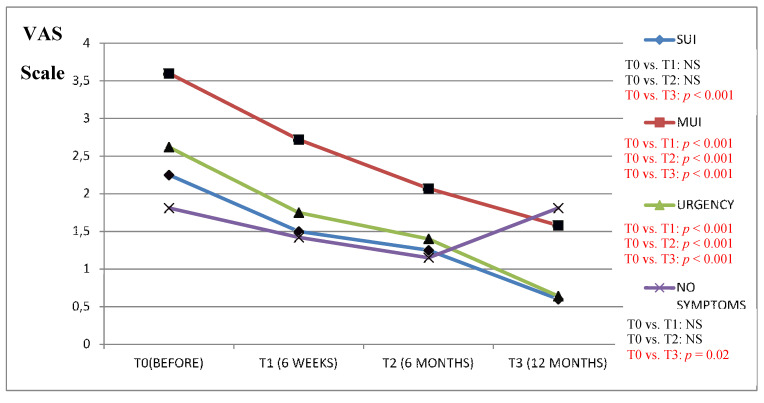
Impact of the vaginal native tissue repair (VNTR) on stool and gas incontinence—according to the visual analogue scale (VAS).

**Figure 4 jcm-09-01634-f004:**
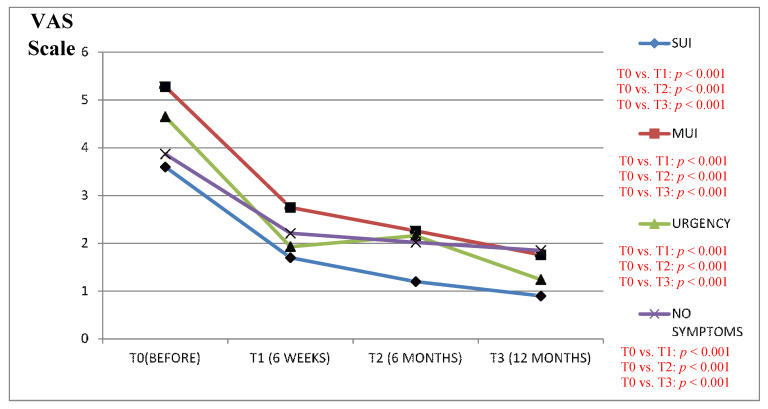
Impact of the vaginal native tissue repair (VNTR) on bulging in everyday functioning—according to the visual analogue scale (VAS).

**Figure 5 jcm-09-01634-f005:**
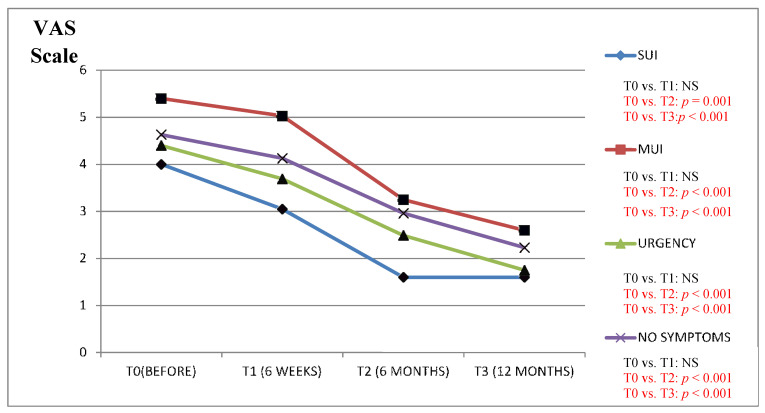
Impact of the vaginal native tissue repair (VNTR) on sexual life—according to the visual analogue scale (VAS).

**Table 1 jcm-09-01634-t001:** Patient demographic data.

Variable	Urgency	SUI	MUI	No LUTS
**Number of Patients**	55	20	72	53
**VD, M ± SD**	2.35 ± 0.95	2.00 ± 1.18	2.59 ± 0.97	2.04 ± 1.00
**CC, M ± SD**	0.27 ± 0.47	0.25 ± 0.5	0.38 ± 0.65	0.17 ± 0.41
**Age (years), M ± SD**	58.69 ± 0.58	54.2 ± 10.43	60.71 ± 8.85	56.34 ± 10.86
**BMI (kg/m^2^), M ± SD**	27.38 ± 3.81	26.10 ± 2.87	28.58 ± 6.94	27.80 ± 4.37
**Hypertension, *n* (%)**	22 (40%)	9 (45%)	35 (48.6%)	25 (47.2%)
**Diabetes, *n* (%)**	7 (12.7%)	2 (10%)	9 (12.5%)	6 (11.3%)
**Hypothroidism, *n* (%)**	5 (9.1%)	1 (5%)	6 (8.3%)	6 (11.3%)
**No Commorbidities, *n* (%)**	9 (16.4%)	3 (15%)	10 (13.9%)	8 (15.1%)

VD: vaginal deliveries, CC: caesarean section, SUI: stress urinary incontinence, MUI: mixed urinary incontinence, LUTS: lower urinary tract symptoms, N: number of patients, M: mean; SD: standard deviation.

**Table 2 jcm-09-01634-t002:** Results of the Prolapse Quality of Life questionnaire (P-QoL).

P-QoL Domains	Study Groups	Before Surgery T0	6 Weeks after T1	6 Months after T2	12 Months after T3	Post Hoc Tukey Test
		M ± SD	M ± SD	M ± SD	M ± SD	
General Health Perceptions	Urgency	49.54 ± 19.72	33.33 ± 19.43	27.31 ± 22.92	16.20 ± 18.91	T0 vs. T1 * T0 vs. T2 * T0 vs. T3 * T1 vs. T2 NS T1 vs. T3 * T2 vs. T3 *
	SUI	57.50 ± 20.03	27.50 ± 19.70	26.25 ± 20.64	11.25 ± 15.12	T0 vs. T1 * T0 vs. T2 * T0 vs. T3 * T1 vs. T2: NS T1 vs. T3 * T2 vs. T3: NS
	MUI	57.50 ± 20.03	27.50 ± 19.70	26.25 ± 20.64	20.71 ± 21.27	T0 vs. T1 * T0 vs. T2 * T0 vs. T3 * T1vs. T2: NS T1 vs. T3 * T2 vs. T3: NS
	N/S	57.50 ± 20.03	27.50 ± 19.70	26.25 ± 20.64	19.71 ± 22.87	T0 vs. T1 * T0 vs. T2 * T0 vs. T3 * T1 vs. T2: NS T1 vs. T3: NS T2 vs. T3 *
Prolapse Impact	Urgency	36.97 ± 31.87	29.09 ± 24.05	24.24 ± 25.22	12.12 ± 22.56	T0 vs. T1: NS T0 vs.T2 * T0 vs. T3 * T1 vs. T2: NS T1 vs. T3 * T2 vs. T3 *
	SUI	41.67 ± 26.21	30.00 ± 26.27	23.33 ± 21.90	10.00 ± 15.67	T0 vs. T1: NS T0 vs.T2: NS T0 vs. T3 * T1 vs. T2: NS T1 vs. T3 * T2 vs. T3: NS
	MUI	45.41 ± 34.29	31.88 ± 27.67	30.43 ± 31.18	18.36 ± 28.32	T0 vs. T1 * T0 vs. T2 * T0 vs. T3 * T1 vs. T2: NS T1 vs. T3 * T2 vs. T3 *
	N/S	46.79 ± 35.09	30.13 ± 28.21	32.69 ± 32.67	13.46 ± 21.14	T0 vs. T1 * T0 vs. T2 * T0 vs. T3 * T1 vs. T2: NS T1 vs. T3 * T2 vs. T3 *
Role Limitations	Urgency	43.40 ± 33.71	32.08 ± 27.51	22.64 ± 28.33	15.09 ± 27.20	T0 vs. T1:NS T0 vs. T2 * T0 vs. T3 * T1 vs. T2: NS T1 vs. T3 * T2 vs. T3: NS
	SUI	40.83 ± 29.85	30.83 ± 26.09	18.33 ± 25.88	6.67 ± 20.52	T0 vs. T1: NS T0 vs. T2 * T0 vs. T3 * T1 vs. T2: NS T1 vs. T3 * T2 vs. T3: NS
	MUI	53.87 ± 35.60	41.55 ± 31.52	33.09 ± 33.64	27.54 ± 34.75	T0 vs. T1 * T0 vs. T2 * T0 vs. T3 * T1 vs. T2: NS T1 vs. T3 * T2 vs. T3: NS
	N/S	37.82 ± 36.32	30.13 ± 32.51	26.92 ± 34.80	23.08 ± 33.84	T0 vs. T1: NS T0 vs. T2 * T0 vs. T3 * T1 vs. T2: NS T1 vs. T3: NS T2 vs. T3: NS
Physical Limitations	Urgency	45.28 ± 34.03	28.30 ± 26.06	23.27 ± 29.84	14.15 ± 24.76	T0 vs. T1 * T0 vs. T2 * T0 vs. T3 * T1 vs. T2: NS T1 vs. T3 * T2 vs. T3: NS
	SUI	44.17 ± 36.38	37.50 ± 33.72	23.33 ± 30.78	8.33 ± 20.59	T0 vs. T1: NS T0 vs. T2: NS T0 vs. T3 * T1 vs. T2: NS T1 vs. T3 * T2 vs. T3: NS
	MUI	59.42 ± 35.41	44.69 ± 32.78	34.54 ± 36.75	36.76 ± 36.76	T0 vs. T1 * T0 vs. T2 * T0 vs. T3 * T1 vs. T2: NS T1 vs. T3 * T2 vs. T3: NS
	N/S	34.62 ± 35.22	27.56 ± 28.94	23.08 ± 30.99	21.47 ± 31.02	T0 vs. T1: NS T0 vs. T2 * T0 vs. T3 * T1 vs. T2: NS T1 vs. T3: NS T2 vs. T3: NS
Social Limitations	Urgency	32.69 ± 28.04	15.17 ± 18.54	11.32 ± 18.73	6.84 ± 14.95	T0 vs. T1 * T0 vs. T2 * T0 vs. T3 * T1 vs. T2: NS T1 vs. T3 * T2 vs. T3: NS
	SUI	30.00 ± 29.97	22.78 ± 31.32	13.33 ± 25.39	6.67 ± 20.52	T0 vs. T1: NS T0 vs. T2: NS T0 vs. T3 * T1 vs. T2: NS T1 vs. T3: NS T2 vs. T3: NS
	MUI	43.30 ± 34.78	27.45 ± 27.85	20.42 ± 29.15	18.95 ± 28.86	T0 vs. T1: NS T0 vs. T2: NS T0 vs. T3: NS T1 vs. T2: NS T1 vs. T3: NS T2 vs. T3: NS
	N/S	21.79 ± 29.35	17.09 ± 24.94	15.81 ± 27.34	16.88 ± 28.31	T0 vs. T1: NS T0 vs. T2: NS T0 vs. T3 * T1 vs. T2: NS T1 vs. T3 * T2 vs. T3 *
Personal Relationships	Urgency	41.51 ± 37.78	36.16 ± 36.51	33.65 ± 38.33	18.55 ± 30.43	T0 vs. T1: NS T0 vs. T2 * T0 vs. T3 * T1 vs. T2 * T1 vs. T3 * T2 vs. T3: NS
	SUI	43.33 ± 36.03	40.83 ± 40.28	19.17 ± 28.75	10.86 ± 26.09	T0 vs. T1: NS T0 vs. T2 * T0 vs. T3 * T1 vs. T2: NS T1 vs. T3 * T2 vs. T3: NS
	MUI	50.25 ± 38.15	43.63 ± 36.88	36.03 ± 39.02	31.13 ± 36.87	T0 vs. T1: NS T0 vs. T2: NS T0 vs. T3 * T1 vs. T2: NS T1 vs. T3: NS T2 vs. T3: NS
	N/S	38.46 ± 36.09	34.29 ± 35.61	29.81 ± 36.65	23.70 ± 34.20	T0 vs. T1 * T0 vs. T2 * T0 vs. T3 * T1 vs. T2: NS T1 vs. T3: NS T2 vs. T3: NS
Emotions	Urgency	51.07 ± 31.91	26.07 ± 27.12	29.06 ± 33.24	19.02 ± 30.67	T0 vs. T1: NS T0 vs. T2 * T0 vs. T3 * T1 vs. T2 * T1 vs. T3 * T2 vs. T3: NS
	SUI	42.22 ± 31.76	35.00 ± 32.50	17.22 ± 25.10	9.44 ± 22.88	T0 vs. T1 * T0 vs. T2 * T0 vs. T3 * T1 vs. T2: NS T1 vs. T3 * T2 vs. T3: NS
	MUI	55.23 ± 34.98	39.05 ± 31.28	31.86 ± 34.97	27.12 ± 34.20	T0 vs. T1 * T0 vs. T2 * T0 vs. T3 * T1 vs. T2: NS T1 vs. T3: NS T2 vs. T3: NS
	N/S	42.09 ± 34.95	30.34 ± 31.67	28.20 ± 33.33	23.50 ± 34.20	T0 vs. T1 * T0 vs. T2 * T0 vs. T3 * T1 vs. T2: NS T1 vs. T3: NS T2 vs. T3: NS
Sleep/Energy	Urgency	36.54 ± 29.34	18.59 ± 23.49	14.74 ± 22.54	10.58 ± 19.53	T0 vs. T1: NS T0 vs. T2 * T0 vs. T3 * T1 vs. T2 * T1 vs. T3 * T2 vs. T3: NS
	SUI	32.50 ± 30.34	23.33 ± 28.31	11.67 ± 24.84	5.83 ± 18.94	T0 vs. T1 * T0 vs. T2 * T0 vs. T3 * T1 vs. T2: NS T1 vs. T3 * T2 vs. T3: NS
	MUI	43.38 ± 31.43	27.70 ± 30.28	20.59 ± 29.24	17.65 ± 27.60	T0 vs. T1 * T0 vs. T2 * T0 vs. T3 * T1 vs. T2: NS T1 vs. T3: NS T2 vs. T3: NS
	N/S	23.40 ± 24.97	13.14 ± 21.73	12.82 ± 22.78	12.50 ± 23.31	T0 vs. T1 * T0 vs. T2 * T0 vs. T3 * T1 vs. T2: NS T1 vs. T3: NS T2 vs. T3: NS
Severity Measures	Urgency	31.45 ± 23.66	7.23 ± 13.38	11.16 ± 20.47	8.49 ± 19.02	T0 vs. T1 * T0 vs. T2 * T0 vs. T3 * T1 vs. T2: NS T1 vs. T3: NS T2 vs. T3: NS
	SUI	37.50 ± 24.56	18.33 ± 20.16	12.08 ± 18.63	7.08 ± 12.47	T0 vs. T1 * T0 vs. T2 * T0 vs. T3 * T1 vs. T2: NS T1 vs. T3: NS T2 vs. T3: NS
	MUI	41.91 ± 31.05	14.71 ± 22.82	16.54 ± 27.56	16.30 ± 27.37	T0 vs. T1 * T0 vs. T2 * T0 vs. T3 * T1 vs. T2: NS T1 vs. T3: NS T2 vs. T3: NS
	N/S	25.32 ± 25.13	11.86 ± 18.70	11.38 ± 19.32	12.02 ± 20.24	T0 vs. T1 * T0 vs. T2 * T0 vs. T3 * T1 vs. T2: NS T1 vs. T3: NS T2 vs. T3: NS

SUI: stress urinary incontinence, MUI: mixed urinary incontinence, N/S—no clinically relevant symptoms from lower urinary tract (LUT). T0: time point 0—before surgery, T1: time point 1—six weeks after surgery, T2: time point 2—six months after surgery, T3: time point 3—twelve months after surgery. * *p <* 0.05 (post hoc Tukey test); M: mean; SD: standard deviation.

**Table 3 jcm-09-01634-t003:** Results of the LUTS questionnaire assessing the severity of the functional disorders of the voiding phase and the post-voiding phase.

		before Surgery	6 Weeks after	6 Months AFTER	12 Months after	Post Hoc Tukey Test
		M ± SD	M ± SD	M ± SD	M ± SD	
Urgency						
Voiding Symptoms	Slow Stream	1.27 ± 1.24	0.71 ± 0.87	0.67 ± 0.69	0.84 ± 0.96	T0 vs. T1 * T0 vs. T2 * T0 vs. T3 * T1 vs. T2: NS T1 vs. T3: NS T2 vs. T3: NS
	Splitting Stream	2.09 ± 1.69	1.13 ± 1.37	1.16 ± 1.54	0.84 ± 1.24	T0 vs. T1 * T0 vs. T2 * T0 vs. T3 * T1 vs. T2: NS. T1 vs. T3: NS T2 vs. T3: NS
	Intermittent Stream	2.0 ± 1.79	1.24 ± 1.53	0.91 ± 1.14	1.00 ± 1.15	T0 vs. T1 * T0 vs. T2 * T0 vs. T3 * T1 vs. T2: NS. T1 vs. T3: NS T2 vs. T3: NS
	Hesistancy	1.51 ± 1.41	0.69 ± 0.79	0.8 ± 1.08	0.91 ± 1.24	T0 vs. T1 * T0 vs. T2 * T0 vs. T3 * T1 vs. T2: NS. T1 vs. T3: NS T2 vs. T3: NS
	Straining	0.98 ± 1.08	0.58 ± 0.57	0.6 ± 0.53	0.60 ± 0.78	T0 vs. T1: NS T0 vs. T2: NS T0 vs. T3: NS T1 vs. T2: NS. T1 vs. T3: NS T2 vs. T3: NS
	Terminal Dribbling	1.34 ± 1.24	0.6 ± 0.89	0.62 ± 0.78	0.65 ± 0.80	T0 vs. T1 * T0 vs. T2 * T0 vs. T3 * T1 vs. T2: NS. T1 vs. T3: NS T2 vs. T3: NS
Post Void Symptoms	Post-Micturition Dribbling	1.22 ± 1.1	0.62 ± 0.87	0.58 ± 0.69	0.65 ± 0.75	T0 vs. T1 * T0 vs. T2 * T0 vs. T3 * T1 vs. T2: NS. T1 vs. T3: NS T2 vs. T3: NS
	Incomplete Emptying	2.04 ± 1.72	1.05 ± 1.43	0.82 ± 1.16	0.65 ± 1.13	T0 vs. T1 * T0 vs. T2 * T0 vs. T3 * T1 vs. T2: NS. T1 vs. T3: NS T2 vs. T3: NS
MUI						
Voiding Symptoms	Slow Stream	1.61 ± 1.27	0.96 ± 1.04	1.13 ± 1.21	1.15 ± 1.23	T0 vs. T1 * T0 vs. T2 * T0 vs. T3 * T1 vs. T2: NS. T1 vs. T3: NS T2 vs. T3: NS
	Splitting Stream	2.42 ± 1.58	1.71 ± 1.53	1.56 ± 1.44	1.47 ± 1.43	T0 vs. T1 * T0 vs. T2 * T0 vs. T3 * T1 vs. T2: NS. T1 vs. T3: NS T2 vs. T3: NS
	Intermittent Stream	2.06 ± 1.61	1.56 ± 1.45	1.63 ± 1.51	1.50 ± 1.43	T0 vs. T1: NS T0 vs. T2: NS T0 vs. T3 * T1 vs. T2: NS. T1 vs. T3: NS T2 vs. T3: NS
	Hesistancy	1.65 ± 1.59	1.24 ± 1.31	1.14 ± 1.28	1.24 ± 1.39	T0 vs. T1: NS T0 vs. T2 * T0 vs. T3: NS T1 vs. T2: NS. T1 vs. T3: NS T2 vs. T3: NS
	Straining	1.53 ± 1.21	0.76 ± 0.96	0.86 ± 1.03	0.86 ± 1.01	T0 vs. T1 * T0 vs. T2 * T0 vs. T3 * T1 vs. T2: NS. T1 vs. T3:NS T2 vs. T3: NS
	Terminal Dribbling	1.57 ± 1.29	0.94 ± 1.02	0.94 ± 1.10	0.92 ± 1.07	T0 vs. T1 * T0 vs. T2 * T0 vs. T3 * T1 vs. T2: NS. T1 vs. T3: NS T2 vs. T3: NS
Post Void Symptoms	Post-Micturition Dribbling	1.26 ± 1.35	0.85 ± 0.97	0.92 ± 1.14	0.88 ± 1.06	T0 vs. T1 * T0 vs. T2: NS T0 vs. T3: NS T1 vs. T2: NS. T1 vs. T3: NS T2 vs. T3: NS
	Incomplete Emptying	2.14 ± 1.71	1.44 ± 1.57	1.75 ± 1.71	1.49 ± 1.62	T0 vs. T1 * T0 vs. T2: NS T0 vs. T3 * T1 vs. T2: NS. T1 vs. T3: NS T2 vs. T3: NS
SUI						
Voiding Symptoms	Slow Stream	0.50 ± 1.05	0.20 ± 0.41	0.30 ± 0.47	0.55 ± 0.51	T0 vs. T1: NS. T0 vs. T2: NS. T0 vs. T2: NS T1 vs. T2: NS. T1 vs. T3: NS T2 vs. T3: NS
	Splitting Stream	1.20 ± 1.51	1.05 ± 1.47	1.05 ± 1.39	0.40 ± 0.50	T0 vs. T1: NS. T0 vs. T2: NS. T0 vs. T3: NS T1 vs. T2: NS. T1 vs. T3: NS T2 vs. T3: NS
	Intermittent Stream	1.30 ± 1.38	0.75 ± 0.97	0.85 ± 1.04	0.90 ± 1.45	T0 vs. T1: NS. T0 vs. T2: NS. T0 vs. T3: NS T1 vs. T2: NS. T1 vs. T3: NS T2 vs. T3: NS
	Hesistancy	1.25 ± 1.33	0.65 ± 0.59	0.65 ± 1.14	0.40 ± 0.50	T0 vs. T1: NS. T0 vs. T2: NS. T0 vs. T3: NS T1 vs. T2: NS. T1 vs. T3: NS T2 vs. T3: NS
	Straining	0.75 ± 0.85	0.40 ± 0.50	0.60 ± 0.50	0.35 ± 0.49	T0 vs. T1: NS. T0 vs. T2: NS. T0 vs. T3: NS T 1 vs. T2: NS. T1 vs. T3: NS T2 vs. T3: NS
	Terminal Dribbling	0.85 ± 0.93	0.50 ± 0.61	0.45 ± 0.60	0.50 ± 0.83	T0 vs. T1: NS. T0 vs. T2: NS. T0 vs. T3: NS T1 vs. T2: NS. T1 vs. T3: NS T2 vs. T3: NS
Post Void Symptoms	Post-Micturition Dribbling	0.90 ± 0.91	0.45 ± 0.76	0.55 ± 0.76	0.70 ± 0.92	T0 vs. T1: NS. T0 vs. T2: NS. T0 vs. T3: NS T1 vs. T2: NS. T1 vs. T3: NS T2 vs. T3: NS
	Incomplete Emptying	1.60 ± 1.19	0.90 ± 1.25	0.50 ± 0.83	0.20 ± 0.41	T0 vs. T1: NS T0 vs. T2 * T0 vs. T3 * T1 vs. T2: NS. T1 vs. T3: NS T2 vs. T3: NS
NO SYMPTOMS						
Voiding Symptoms	Slow Stream	0.47 ± 0.77	0.45 ± 0.64	0.58 ± 0.63	0.58 ± 0.69	T0 vs. T1: NS. T0 vs. T2: NS. T0 vs. T3: NS T1 vs. T2: NS. T1 vs. T3: NS T2 vs. T3: NS
	Splitting Stream	1.21 ± 1.65	1.25 ± 1.54	1.32 ± 1.55	1.32 ± 1.45	T0 vs. T1: NS. T0 vs. T2: NS. T0 vs. T3: NS T1 vs. T2: NS. T1 vs. T3: NS T2 vs. T3: NS
	Intermittent Stream	1.17 ± 1.30	0.91 ± 1.15	1.08 ± 1.33	1.04 ± 1.44	T0 vs. T1: NS. T0 vs. T2: NS. T0 vs. T3: NS T1 vs. T2: NS. T1 vs. T3: NS T2 vs. T3: NS
	Hesistancy	0.77 ± 1.01	0.74 ± 1.06	0.68 ± 0.96	0.70 ± 1.01	T0 vs. T1: NS. T0 vs. T2: NS. T0 vs. T3: NS T1 vs. T2: NS. T1 vs. T3: NS T2 vs. T3: NS
	Straining	0.51 ± 0.70	0.51 ± 0.61	0.74 ± 0.76	0.72 ± 0.79	T0 vs. T1: NS. T0 vs. T2: NS. T0 vs. T3: NS T1 vs. T2: NS. T1 vs. T3: NS T2 vs. T3: NS
	Terminal Dribbling	0.68 ± 0.85	0.72 ± 0.86	0.58 ± 0.75	0.68 ± 0.87	T0 vs. T1: NS. T0 vs. T2: NS. T0 vs.T3: NS T1 vs. T2: NS. T1 vs. T3: NS T2 vs. T3: NS
Post Void Symptoms	Post-Micturition Dribbling	0.74 ± 0.88	0.64 ± 0.88	0.55 ± 0.75	0.68 ± 0.83	T0 vs. T1: NS. T0 vs. T2: NS. T0 vs. T3: NS T1 vs. T2: NS. T1 vs. T3: NS T2 vs. T3: NS
	Incomplete Emptying	1.11 ± 1.46	0.60 ± 1.13	0.72 ± 1.23	0.70 ± 1.20	T0 vs. T1 * T0 vs. T2: NS T0 vs. T3: NS T1 vs. T2: NS. T1 vs. T3: NS T2 vs. T3: NS

SUI: stress urinary incontinence; MUI: mixed urinary incontinence; T0: time point 0—before surgery; T1: -time point 1—six weeks after surgery; T2: time point 2—six months after surgery; T3: time point 3—twelve months after surgery; * *p <* 0.05 (post hoc Tukey test); M: mean; SD: standard deviation.

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
