# Peer review of "The Influence of Vaginal Native Tissue Repair (VNTR) on Various Aspects of Quality of Life in Women with Symptomatic Pelvic Organ Prolapse—A Prospective Cohort Study"

_jcm, 2020, doi:10.3390/jcm9061634_

Round 1
Reviewer 1 Report
The article studied important question, how different urinary problems get relieved by native tissue repair.
First problem is that there was not any power analysis to show how big groups should have been. Moreover, there was a big difference in numbers of the women between the groups. However, the study showed that women suffering from urge got most benefit after 12 months.
You should state the design of your study, prospective cohort study.
Throughout the manuscript is would be clearer to use urgency, SUI, MUI etc instead of groups 1 to 4, as well as not to use T0, but instead use the terms before surgery, after..
Many sentences are complicated and difficult to read, the language should be revised throughout the manuscript.
Be consistent, state in same order first VAS and then P-QoL or vice versa, but always in same order.
Abstract:
State more clearly in which VAS the benefit was gain.
Introduction:
lines 40-41, do not use paracentesis, include the information to the sentence or leave it out.
lines 41-44, the association between ultrasound diagnostic of levator ani lesion and risk of prolapse is stated really complicated, re-write this.
lines 44-46: this is not correlation, it is association.
You could state that VNTR is the first choice for POP in most of the cases.
M & M
Was this the first operation for the women? This is stated in the end of the discussion, state it here.
Line 87, insert the reference.
Results:
The results are hard to read and follow.
You could show in each group how many had very mild, mild, moderate, strong, or very strong symptoms.
You need to show the POP state, it would be interesting to know if the women with urge or MUI had more advanced prolapse etc.
Include some more basic demographic data to table 1, at least those above, chronic illnesses etc. The table would be more clear vice versa; groups in X-axis.
Tables 2 and 3 are impossible to read! You can`t give all that information, leave both ANOVAN and post hoc boxes out, indicate e.g. by asterisk those which are significant.
You could state in the text that the women with MUI had most bother in almost all the studied dimensions.
Be logic, first short-term results and then long-term.
Why are figures 2a and 2b not in the same figure?
Line 149 > it is really difficult to follow T0 to … use text that the reader does not need to remember your codes.
Discussion:
Discussion is too long and not consistent, should be re-written.
Start the discussion by stating your main results. The two first paragraphs are not related to the aim of your study.
line 203: I think you mean clinical instead of statistical. It is not stated whether you operated only women with cystocele or also those with rectosele, but here you are discussing on posterior compartment.
You should clearly state in the results, e.g. in short term, all women with any urinary problem got benefit in General… and Severity.. and then long-term, where after you should discuss why the results differed between the groups. And after that compare to other studies.
Lines 228, the discussion above on the effect of VNTR on SUI is true, but the conclusion is not, in addition to surgical technique the pathophysiology behind likely mostly explain the contradictory results.
Combine the discussion on the groups 1 and 3, since they show similar results, and this confirms the previous data that urge is regarded subjectively the most distressful.
Line 243-245 leave out, state in M&M how the women were followed and that they had a possibility to contact the operator.
Line 248 > how is the following discussion related to the present data (importance of apical support)?
line 260 you are comparing your results to those of prostethic surgery, but your reference (27) is native tissue repair. Moreover, you have not studied in this data the complications, so why to discuss them here?
In your conclusion you should note that the data is short-term and these women should be followed for long-term data.
Author Response
Dear Reviewers,
Thank you for your careful reading of our manuscript. We really appreciate your helpful comments and suggestions. We tried our best in order to follow all valuable remarks.
Reviewer #1:
1.The article studied important question, how different urinary problems get relieved by native tissue repair.
First problem is that there was not any power analysis to show how big groups should have been. Moreover, there was a big difference in numbers of the women between the groups. However, the study showed that women suffering from urge got most benefit after 12 months.
We absolutely agree with Your suggestion. Below please find a power calculation and sample size required. We added this information to the body text together with specific citation.
The statistical required sample size for effect size f=0.25 (medium effect size convention) and 0.8 power of the study was computed using the ANOVA a priori test (G*Power, Düsseldorf, Germany)[1]. The size of the sample was estimated at 180.
- Faul, F., Erdfelder, E., Lang, A.-G., & Buchner, A. (2007). G*Power 3: A flexible statistical power analysis program for the social, behavioral, and biomedical sciences. Behavior Research Methods, 39, 175-191
You should state the design of your study, prospective cohort study.
Thank you for your comment. This was prospective cohort study. The subsequent changes have been made according to reviewer’s suggestion and the manuscript has been rewritten.
- Throughout the manuscript is would be clearer to use urgency, SUI, MUI etc instead of groups 1 to 4, as well as not to use T0, but instead use the terms before surgery, after..
Thank you for your comment. The subsequent changes have been made according to reviewer’s suggestion and the manuscript has been rewritten. We strictly followed your suggestion and removed all descriptions (T0, T1, T2, T3) from the text and from the tables as well, however if you agree we suggest to keep these descriptions in the tables because when we removed those descriptions (see below) such busy table is very difficult to follow. These descriptions in the table are easy to follow because reader has this information in front of his eyes and therefore the statistical analysis is easier to follow. Below we present the Table without these description and we feel that it is really confusing.
Table 2. Results of Prolapse Quality of Life questionnaire (P-QoL)
|
P-QoL domains |
Study groups |
before surgery |
6 weeks after |
6 months after |
12 months after
|
|
M±SD |
M±SD |
M±SD |
M±SD |
||
|
General Health Perceptions |
Urgency |
49.54±19.72 |
33.33±19.43* |
27.31±22.92’~ |
16.20±18.91”^ |
|
SUI |
57.50±20.03 |
27.50±19.70* |
26.25±20.64’ |
11.25±15.12”^ |
|
|
MUI |
57.50±20.03 |
27.50±19.70* |
26.25±20.64’ |
20.71±21.27”^ |
|
|
N/S |
57.50±20.03 |
27.50±19.70* |
26.25±20.64’~ |
19.71±22.87” |
|
|
Prolapse Impact |
Urgency |
36.97±31.87 |
29.09±24.05 |
24.24±25.22’~ |
12.12±22.56”^ |
|
SUI |
41.67±26.21 |
30.00±26.27 |
23.33±21.90 |
10.00±15.67”^ |
|
|
MUI |
45.41±34.29 |
31.88±27.67* |
30.43±31.18’~ |
18.36±28.32”^ |
|
|
N/S |
46.79±35.09 |
30.13±28.21* |
32.69±32.67’~ |
13.46±21.14”^ |
|
|
Role Limitations |
Urgency |
43.40±33.71 |
32.08±27.51 |
22.64±28.33’ |
15.09±27.20”^ |
|
SUI |
40.83±29.85 |
30.83±26.09 |
18.33±25.88’ |
6.67±20.52”^ |
|
|
MUI |
53.87±35.60 |
41.55±31.52* |
33.09±33.64’ |
27.54±34.75”^ |
|
|
N/S |
37.82±36.32 |
30.13±32.51 |
26.92±34.80’ |
23.08±33.84” |
|
|
Physical Limitations |
Urgency |
45.28±34.03 |
28.30±26.06* |
23.27±29.84’ |
14.15±24.76”^ |
|
SUI |
44.17±36.38 |
37.50±33.72 |
23.33±30.78 |
8.33±20.59”^ |
|
|
MUI |
59.42±35.41 |
44.69±32.78* |
34.54±36.75’ |
36.76±36.76”^ |
|
|
N/S |
34.62±35.22 |
27.56±28.94 |
23.08±30.99’ |
21.47±31.02” |
|
|
Social Limitations |
Urgency |
32.69±28.04 |
15.17±18.54* |
11.32±18.73’ |
6.84±14.95”^ |
|
SUI |
30.00±29.97 |
22.78±31.32 |
13.33±25.39 |
6.67±20.52” |
|
|
MUI |
43.30±34.78 |
27.45±27.85 |
20.42±29.15 |
18.95±28.86 |
|
|
N/S |
21.79±29.35 |
17.09±24.94 |
15.81±27.34~ |
16.88±28.31”^ |
|
|
Personal Relationships |
Urgency |
41.51±37.78 |
36.16±36.51 |
33.65±38.33’ + |
18.55±30.43”^ |
|
SUI |
43.33±36.03 |
40.83±40.28 |
19.17±28.75’ |
10.86±26.09”^ |
|
|
MUI |
50.25±38.15 |
43.63±36.88 |
36.03±39.02 |
31.13±36.87” |
|
|
N/S |
38.46±36.09 |
34.29±35.61* |
29.81±36.65’ |
23.70±34.20” |
|
|
Emotions |
Urgency |
51.07±31.91 |
26.07±27.12 |
29.06±33.24’ + |
19.02±30.67”^ |
|
SUI |
42.22±31.76 |
35.00±32.50* |
17.22±25.10’ |
9.44±22.88”^ |
|
|
MUI |
55.23±34.98 |
39.05±31.28* |
31.86±34.97’ |
27.12±34.20” |
|
|
N/S |
42.09±34.95 |
30.34±31.67* |
28.20±33.33’ |
23.50±34.20” |
|
|
Sleep/Energy |
Urgency |
36.54±29.34 |
18.59±23.49 |
14.74±22.54’ + |
10.58±19.53”^ |
|
SUI |
32.50±30.34 |
23.33±28.31* |
11.67±24.84’ |
5.83±18.94”^ |
|
|
MUI |
43.38±31.43 |
27.70±30.28* |
20.59±29.24’ |
17.65±27.60” |
|
|
N/S |
23.40±24.97 |
13.14±21.73* |
12.82±22.78’ |
12.50±23.31” |
|
|
Severity Measures |
Urgency |
31.45±23.66 |
7.23±13.38* |
11.16±20.47’ |
8.49±19.02” |
|
SUI |
37.50±24.56 |
18.33±20.16* |
12.08±18.63’ |
7.08±12.47” |
|
|
MUI |
41.91±31.05 |
14.71±22.82* |
16.54±27.56’ |
16.30±27.37” |
|
|
N/S |
25.32±25.13 |
11.86±18.70* |
11.38±19.32’ |
12.02±20.24” |
SUI - stress urinary incontinence, MUI - mixed urinary incontinence, N/S – no clinically relevant symptoms from LUT. *p<0.05 between time point before surgery and 6 weeks after; ‘p<0.05 between timepoint before surgery and 6 months after; “p<0.05 between time point before surgery and 12 months after; + p<0.05 between time point 6 weeks after surgery and 6 months after; ~p<0.05 between time point 6 months after surgery and 12 months after;^ p<0.05 between time point 6 weeks after surgery and 12 months after
- Many sentences are complicated and difficult to read, the language should be revised throughout the manuscript.
Thank you for your comment. The subsequent changes have been made according to reviewer’s suggestion and the manuscript has been rewritten.
- Be consistent, state in same order first VAS and then P-QoL or vice versa, but always in same order.
Thank you for your comment. The subsequent changes have been made according to reviewer’s suggestion and the manuscript has been rewritten. We strictly followed your suggestions and presented first Pqol followed by VAS scale results.
- Abstract: State more clearly in which VAS the benefit was gain.
Thank you for your comment. In abstract we clearly stated that significant changes in VAS scale were found in all symptoms in Urgency and MUI groups. We were unable in the abstract describe in details changes in other groups because of word limit for the abstract (200 words). Currently we have 199 words.
- Introduction:
lines 40-41, do not use paracentesis, include the information to the sentence or leave it out.
Thank you for your comment. Paracentesis has been removed.
- lines 41-44, the association between ultrasound diagnostic of levator ani lesion and risk of prolapse is stated really complicated, re-write this.
Thank you for your comment. The subsequent changes have been made according to reviewer’s suggestion and the manuscript has been rewritten.
There is sufficient evidence to state that delivery-related levator trauma is associated with objective prolapse on clinical assessment especially of the anterior and central compartments, with
an odds ratio of 5.7 (95% CI, 3.5–9.1).
- lines 44-46: this is not correlation, it is association.
Thank you for your comment. The subsequent changes have been made according to reviewer’s suggestion and the manuscript has been rewritten.
- You could state that VNTR is the first choice for POP in most of the cases.
Thank you for your comment. We changed the text accordingly.
- M & M
Was this the first operation for the women? This is stated in the end of the discussion, state it here.
Thank you for your comment. That was the patients first POP reconstructive surgery and we added this information to the text.
We inserted this information in M& M section
- Line 87, insert the reference.
Thank you for your comment. We added the necessary information to the text.
Guldberg R, Kesmodel US, Hansen JK, Gradel KO, Brostrøm S, Kærlev L, Nørgård BM. Patient Reported Outcome Measures in Women Undergoing Surgery for Urinary Incontinence and Pelvic Organ Prolapse in Denmark, 2006-2011. Int Urogynecol J 2013 Jul;24(7):1127-34. doi: 10.1007/s00192-012-1979-5.
- Results:
The results are hard to read and follow.
- You could show in each group how many had very mild, mild, moderate, strong, or very strong symptoms.
Thank you for your comment. This information was reported previously together with clinical advancement of prolapse according to POP scale.
The POP-Q assessment revealed 58 (29%) women at stage II, 117 (58.5%) at stage III, and 25 (12.5%) at stage IV without significant differences in distribution among study groups. We added this information to the body text.
Rechberger, E.; Skorupska, K.; Rechberger, T.; Wojtaś, M.; Miotła, P.; Kulik-Rechberger, B.; Wróbel, A. The Influence of Vaginal Native Tissues Pelvic Floor Reconstructive Surgery in Patients with Symptomatic Pelvic Organ Prolapse on Preexisting Storage Lower Urinary Tract Symptoms (LUTS). J Clin Med 2020, 18, 9. 829; DOI: 10.3390/jcm9030829.
- You need to show the POP state, it would be interesting to know if the women with urge or MUI had more advanced prolapse etc.
Thank you for your comment. Please see above.
We added this information as suggested
- Include some more basic demographic data to table 1, at least those above, chronic illnesses etc. The table would be more clear vice versa; groups in X-axis.
Thank you for your comment Table 1 was modified and the missing information has been added.
- Tables 2 and 3 are impossible to read! You can`t give all that information, leave both ANOVAN and post hoc boxes out, indicate e.g. by asterisk those which are significant.
Thank you for your comment. The subsequent changes have been made according to reviewer’s suggestion and the manuscript has been rewritten. Please find our explanation above- remark no 3.
- You could state in the text that the women with MUI had most bother in almost all the studied dimensions.
Thank you for your comment. The subsequent changes have been made according to reviewer’s suggestion and the manuscript has been rewritten.
- Be logic, first short-term results and then long-term.
Thank you for your comment. In fact 12 months observation time is considered as short term observation. Therefore the long term was replaced by 12 months in order to be consistent. Six weeks after surgery was considered as postoperative period.
- Why are figures 2a and 2b not in the same figure?
Thank you for your comment. On figure 2a we presented the influence of UI (SUI and MUI) severity on everyday functioning whereas on figure 2b the influence of urgency on everyday functioning. We don’t use term OAB wet and OAB dry in order to be consistent with ICS terminology where OAB is the presence of symptoms without any other obvious pathology including POP. Since all our patients were operated due to symptomatic prolapse we couldn’t use the term OAB neither wet nor dry.
- Line 149 > it is really difficult to follow T0 to … use text that the reader does not need to remember your codes.
Thank you for your comment. We changed the body text accordingly.
- Discussion:
Discussion is too long and not consistent, should be re-written.
Thank you for your comment. Discussion has been rewritten.
- Start the discussion by stating your main results. The two first paragraphs are not related to the aim of your study.
Thank you for this suggestion We started the discussion with short summary of our results.
- line 203: I think you mean clinical instead of statistical. It is not stated whether you operated only women with cystocele or also those with rectosele, but here you are discussing on posterior compartment.
Thank you for your comment . All our patients underwent also posterior compartment restoration
- You should clearly state in the results, e.g. in short term, all women with any urinary problem got benefit in General… and Severity.. and then long-term, where after you should discuss why the results differed between the groups. And after that compare to other studies.
Thank you for this suggestion. In fact this is short term study. The observation time is 12 months. Therefore we change the text and instead to use short-term results we used description 6 weeks after surgery.
- Lines 228, the discussion above on the effect of VNTR on SUI is true, but the conclusion is not, in addition to surgical technique the pathophysiology behind likely mostly explain the contradictory results.
We absolutely agree with Your comments and we added this information to the text
- Combine the discussion on the groups 1 and 3, since they show similar results, and this confirms the previous data that urge is regarded subjectively the most distressful.
Thank you for your comment. The subsequent changes have been made according to reviewer’s suggestion and the manuscript has been rewritten.
- Line 243-245 leave out, state in M&M how the women were followed and that they had a possibility to contact the operator.
Thank you for your comment. The subsequent changes have been made according to reviewer’s suggestion and the manuscript has been rewritten.
- Line 248 > how is the following discussion related to the present data (importance of apical support)?
Thank you for your comment . Yes all patients in our study group underwent apical repair and this is the reason that we included this information in discussion .
- line 260 you are comparing your results to those of prostethic surgery, but your reference (27) is native tissue repair. Moreover, you have not studied in this data the complications, so why to discuss them here?
Thank You for this comment. We removed this part of discussion together with reference 27.
We changed the discussion in order to address all above mentioned suggestions
- In your conclusion you should note that the data is short-term and these women should be followed for long-term data.
Thank you for your comment . We absolutely agree and this sentence was added to our conclusions.
Reviewer 2 Report
is an interesting article that shows data related to native tissue vaginal surgery. obviously after the FDA warning there was an increase in the use of this technique
Abstarct: exhaustive.
Introduction: well written and the topic is well explained.
Methods: well written,
Results: exhaustive and the tables are easy to interpret.
Discussion: well explained but I have only a few considerations:
discuss more about the symptoms of posterior compartment and the effects on quality of life and sexual function after vaginal native tissue repair
remember how fascial surgery without using mesh is very effective even in rectocele……
Author Response
Dear Reviewers,
Thank you for your careful reading of our manuscript. We really appreciate your helpful comments and suggestions. We tried our best in order to follow all valuable remarks.
Reviewer #2:
1.discuss more about the symptoms of posterior compartment and the effects on quality of life and sexual function after vaginal native tissue repair
Thank you for your comment . The subsequent changes have been made according to reviewer’s suggestion and the manuscript has been rewritten.
- remember how fascial surgery without using mesh is very effective even in rectocele……
Thank you for these comments. We included these information in discussion.
Reviewer 3 Report
The present study evaluates the effect of vaginal native tissue repair (VNTR) on quality of life.
This is a very nice and elegant study. I have some comments and questions
- Please specify if this was retrospective or prospective study
- I am somehow perplexed why there is no urgency incontinence group? If it is included in “urgency” it should be defined as “urgency with or without incontinence” or OAB wet or dry
- Group “No LUTS”. Why did these patients have surgery? What symptoms did they have? How can you evaluate voiding symptoms in this group if they had no LUTS symptoms?
- In Table 3, in columns T0-T3, what does the numbers in rows under M±SD mean? Are theses numbers the last number of SD and are below because the columns are too narrow. This is confusing and must be corrected.
- The conclusion should correlate with the aim and title of the study and should read “VNTR is an effective, safe and durable surgical method for improving symptoms and quality of life of the patients with symptomatic pelvic organ prolapse.
The present study evaluates the effect of vaginal native tissue repair (VNTR) on quality of life.
Author Response
Dear Reviewers,
Thank you for your careful reading of our manuscript. We really appreciate your helpful comments and suggestions. We tried our best in order to follow all valuable remarks.
Reviewer #3:
1.Please specify if this was retrospective or prospective study
Thank you for your comment. The subsequent changes have been made according to reviewer’s suggestion and the manuscript has been rewritten.
2. I am somehow perplexed why there is no urgency incontinence group? If it is included in “urgency” it should be defined as “urgency with or without incontinence” or OAB wet or dry
Thank you for your comment. Urgency group consists of both - urgency with or without incontinence. We don’t use term OAB wet and OAB dry in order to be consistent with ICS terminology where OAB is the presence of symptoms without any other obvious pathology including POP. Since all our patients were operated due to symptomatic prolapse we couldn’t use the term OAB neither wet nor dry.
- Group “No LUTS”. Why did these patients have surgery? What symptoms did they have? How can you evaluate voiding symptoms in this group if they had no LUTS symptoms?
Thank you for your comment. Those patients had surgery due to symptomatic prolapse but they did not have any concomitant LUT symptoms.
4.In Table 3, in columns T0-T3, what does the numbers in rows under M±SD mean? Are theses numbers the last number of SD and are below because the columns are too narrow. This is confusing and must be corrected.
Thank you for your comment. The subsequent changes have been made according to reviewer’s suggestion and the manuscript has been rewritten.
- The conclusion should correlate with the aim and title of the study and should read “VNTR is an effective, safe and durable surgical method for improving symptoms and quality of life of the patients with symptomatic pelvic organ prolapse. The present study evaluates the effect of vaginal native tissue repair (VNTR) on quality of life.
Thank you for your comment. The subsequent changes have been made according to reviewer’s suggestion and the manuscript has been rewritten.
Round 2
Reviewer 1 Report
Thank you for the major revision of the manuscript.